# A High-Homology Region Provides the Possibility of Detecting β-Barrel Pore-Forming Toxins from Various Bacterial Species

**DOI:** 10.3390/ijms25105327

**Published:** 2024-05-14

**Authors:** Alexey S. Nagel, Olesya S. Vetrova, Natalia V. Rudenko, Anna P. Karatovskaya, Anna V. Zamyatina, Zhanna I. Andreeva-Kovalevskaya, Vadim I. Salyamov, Nadezhda A. Egorova, Alexander V. Siunov, Tatiana D. Ivanova, Khanafi M. Boziev, Fedor A. Brovko, Alexander S. Solonin

**Affiliations:** 1FSBIS FRC Pushchino Scientific Centre of Biological Research, G.K. Skryabin Institute of Biochemistry and Physiology of Microorganisms, Russian Academy of Sciences, 5 Prospekt Nauki, 142290 Pushchino, Moscow Region, Russia; anagell@mail.ru (A.S.N.); hemolysin6@gmail.com (Z.I.A.-K.); v.salyamoff@yandex.ru (V.I.S.); av_siunov@rambler.ru (A.V.S.); tan4ikovaya@mail.ru (T.D.I.); solonin.a.s@yandex.ru (A.S.S.); 2Pushchino Branch, Shemyakin–Ovchinnikov Institute of Bioorganic Chemistry, Russian Academy of Sciences, 6 Prospekt Nauki, 142290 Pushchino, Moscow Region, Russia; olesja.wetrowa1999@gmail.com (O.S.V.); annakaratovskaya@mail.ru (A.P.K.); anna.zamjatina@yandex.ru (A.V.Z.); bozievkh@mail.ru (K.M.B.); brovko@bibch.ru (F.A.B.); 3Federal State Budgetary Educational Institution of Higher Education “Ryazan State University Named for S.A. Yesenin”, 46 st. Svobody, 390000 Ryazan, Ryazan Region, Russia; nickase@mail.ru

**Keywords:** *Staphylococcus aureus* α-toxin, *Bacillus cereus* hemolysin II, cytotoxin K2, suppression of hemolytic activity, monoclonal antibodies, enzyme immunoassay

## Abstract

The pathogenicity of many bacteria, including *Bacillus cereus* and *Staphylococcus aureus*, depends on pore-forming toxins (PFTs), which cause the lysis of host cells by forming pores in the membranes of eukaryotic cells. Bioinformatic analysis revealed a region homologous to the Lys171-Gly250 sequence in hemolysin II (HlyII) from *B. cereus* in over 600 PFTs, which we designated as a “homologous peptide”. Three β-barrel PFTs were used for a detailed comparative analysis. Two of them—HlyII and cytotoxin K2 (CytK2)—are synthesized in *Bacillus cereus sensu lato*; the third, *S. aureus* α-toxin (Hla), is the most investigated representative of the family. Protein modeling showed certain amino acids of the homologous peptide to be located on the surface of the monomeric forms of these β-barrel PFTs. We obtained monoclonal antibodies against both a cloned homologous peptide and a 14-membered synthetic peptide, DSFNTFYGNQLFMK, as part of the homologous peptide. The HlyII, CytK2, and Hla regions recognized by the obtained antibodies, as well as an antibody capable of suppressing the hemolytic activity of CytK2, were identified in the course of this work. Antibodies capable of recognizing PFTs of various origins can be useful tools for both identification and suppression of the cytolytic activity of PFTs.

## 1. Introduction

The plasma membrane plays an extremely important role in the existence of any cell, as it separates its contents from the outer environment, determines the regulation of the transfer of substances into and out of the cell to ensure its integrity, and regulates the exchange between the cell and the environment. Disturbance of the membrane’s barrier function can lead to the death of target cells [1]. Pathogenic bacteria produce substances that directly damage structures or kill cells of the macroorganism, facilitating their penetration into the eukaryotic cell. These agents play major roles in the development of diseases caused by bacteria. An important role in the pathogenesis of such substances is played by cytolytic pore-forming toxins (PFTs). Almost all pathogenic bacteria are capable of disrupting the permeability of cell membranes via specialized PFTs to form oligomeric pores in the membrane. These toxins are the most common bacterial cytotoxic proteins necessary for the virulence of a variety of bacterial pathogens and for promoting the survival of pathogenic bacteria in the transition of these bacteria to the eukaryotic environment [2]. The formation of pores causes the swelling of cells, followed by their death. One or two pores per cell can cause the complete destruction of eukaryotic cells [3]. PFTs are usually synthesized as water-soluble molecules that penetrate the membrane in monomeric form and are formed into oligomeric pores within the membrane upon interaction with the membranes of target cells [4,5]. Despite the slight similarity of PFTs’ primary sequences, one of the families of these proteins, namely, β-barrel PFTs, demonstrates not only the conformational similarity, but also the universality of the functional mechanisms of action of these toxins [6]. Comparison of the amino acid sequences of β-PFTs with the *Bacillus cereus* Lys171-Gly250 hemolysin II (HlyII) sequence revealed, in their primary structures, a peptide region with a high degree of homology to this peptide that we called a “homologous peptide” (HP), which contains the identical sequence 219YGNQLFM225 (according to the HlyII sequence) [7]. An effective way to suppress cytolytic activity is to use monoclonal antibodies (mAbs) against certain regions of PFTs’ amino acid sequences [8]. This has been demonstrated for *Staphylococcus aureus* α-toxin (Hla) [9], *Streptococcus pneumoniae* pneumolysin [10], and *B. cereus* HlyII [11], among others. The aim of this work was to obtain mAbs against the homologous peptide of *B. cereus* HlyII as a tool for the simultaneous identification and assessment of the level of expression of various β-PFTs. We hypothesize that the obtained mAbs against the HP will enable the identification of a number of β-barrel PFT representatives from various species of pathogenic bacteria. They may also be able to specifically detect the presence of water-soluble forms and inhibit the cytolytic activity of certain β-barrel PFTs.

## 2. Results

### 2.1. Selection of a Peptide Region to Produce mAbs against a Number of Bacterial PFTs

Using the HMMER tool [12], we analyzed the sequence of the HP for the presence of groups of identical and similar amino acid residues among the proteins using the UniProtKB database [13]. The results of the analysis showed regions similar to the HP among representatives of different families of microorganisms. Figure 1 shows the representativeness of those microorganisms with β-barrel PFTs that contain regions that are homologous to the HlyII Lys171-Gly250 sequence (HP), with varying degrees of similarity. This group includes representatives of the following families: *Bacillaceae* (389), *Staphilococcaceae* (181), *Paenibacillaceae* (8), *Enterococcaceae* (1), *Clostridiaceae* (36), *Pseudomonadaceae* (2), and *Sphingomonadaceae* (1). The number of hits for the HP among the representatives of these families is indicated in parentheses. Additionally, the same homologous regions were found in 30 representatives of *Caudovirales* viruses and three streptococcal phages. Thus, bioinformatic analysis of the UniProtKB amino acid sequences revealed more than 600 β-barrel PFTs with a high-homology region in their composition. Figure 2 shows that a sequence with a significant homology is present in the HlyII and cytotoxin K2 (CytK2) of *B. cereus*, in the Hla of *S. aureus*, in the necrotic enterotoxins of *Clostridium perfringens*, in the β-pore-forming cytolysins of *C. septicum* and *C. botulinum*, and several others.

Production and purification of the HP were achieved by cloning this region into an expression vector (see Section 4). To increase immunogenicity, the sequence of the HP was expanded in the N-terminal part to the region involved in the formation of the oligomeric spatial structure of the stem part or transmembrane channel and the triangle region corresponding to the PFT region inserted into the membrane [14]. The HP at the C-terminus includes the HlyII Met225-Gly250 region, which is a part of the HlyIILCTD described previously [15] (Figure 3). Additionally, to obtain another series of mAbs, we used the synthesized oligopeptide (SHP). This 14-membered synthetic peptide included a primary sequence of seven amino acid residues—219YGNQLFM225, identical to several PFTs—that are part of the HP.

The amino acid sequence of the HP contained more than 40% identical amino acid residues to a number of proteins of the β-PFT family. The degree of similarity in this region with Hla of *S. aureus* and CytK2 of *B. cereus* was 47%. For comparison, we selected three β-barrel PFTs, two of which are synthesized in *B. cereus sensu lato* cells—HlyII and CytK2; the third protein chosen for analysis is the most studied representative—Hla from *S. aureus*. Protein modeling showed that some of the amino acids that make up the HP (Figure 3 and Figure 4) are located on the surface of the monomeric forms of these PFTs (Figure 4). The triangle region and the stem strands are formed by the most conserved sections of the amino acid sequence, characterized by high hydrophobicity and smaller side radicals. These regions play an important role in conformational rearrangements during oligomerization and are also involved in interactions between monomers in the transition of the water-soluble form of the toxin to the membrane-bound form [16,17].

### 2.2. Production of mAbs against the HP and SHP

A preparation of the recombinant HP was used as an antigen to produce HP-series mAbs. During immunization, the immune response of animals was assessed by solid-phase ELISA for interaction with the immobilized HP and recombinant *B. cereus* HlyII and *S. aureus* Hla preparations. The maximum dilution of immune serum at a dose of 20 μg/mouse of the administered antigen upon interaction with HlyII was 1/128,000 (titer); with Hla, it was 1/16,000. Splenocytes from this animal were used as a source of lymphocytes for hybridomas secreting mAbs against the homologous peptide according to the method of Keller and Milstein [20].

To obtain SHP-series mAbs, the SHP was conjugated with keyhole limpet hemocyanin (KLH) using glutaraldehyde as a cross-linking agent. To produce hybridomas secreting mAbs against the synthetic peptide, we used mouse splenocytes, the serum of which interacted with the immobilized antigen to the greatest extent, up to a dilution of 1/64,000.

Based on assessment of the proliferative activity and stability of antibody production, we selected five stable hybridoma clones, secreting anti-HlyII HP-series mAbs, obtained after immunization with the homologous peptide. HP-1, HP-3, HP-4, and HP-7 contained the κ light chain and the γ1 heavy chain; HP-5 contained Igµλ. We also obtained four stable hybridoma clones secreting anti-HlyII mAbs of the SHP series after immunization with the synthetic peptide, namely, SHP-1, SHP-2, SHP-3, and SHP-4, each of which contained the µ heavy chain. SHP-1 and SHP-4 contained the κ light chain, while SHP-2 and SHP-3 contained the λ light chain.

### 2.3. Antigen-Binding Activity of the Resulting mAbs

The antigen-binding activity of the resulting mAbs was studied by solid-phase ELISA under equal antigen sorption conditions and by immunoblotting. HP-1, HP-3, and HP-7 antibodies were found among the HP-series antibodies that can effectively recognize *B. cereus* HlyII but cannot interact with HlyIILCTD (Met225-Ile412) or with *S. aureus* Hla and *B. cereus* CytK2 (Figure 5a). These antibodies recognize the HlyII region within Lys171-Arg212, which does not contain an identical sequence of amino acid residues between PFTs. This conclusion was confirmed by immunoblotting (Figure 5b) and the results presented below.

Figure 5a shows that antibody HP-4 interacts effectively with HlyIILCTD, which indicates that its binding region is at the C-terminal part of HP within the HlyII Met225-Gly250 sequence.

ELISA demonstrated that all of the resulting HP-series mAbs (Figure 5a) recognized the HP in the full-length HlyII of *B. cereus*. All of the obtained SHP-series mAbs effectively recognized HlyII, CytK2 of *B. cereus*, and Hla of *S. aureus* (Figure 6a,b). These results confirm that the DSFNTFYGNQLFMK sequence from *B. cereus* HlyII, containing the primary YGNQLFM sequence identical in these PFTs, provides a tool (mAbs) for the simultaneous identification of at least three β-pore-forming toxins.

HP-5 and SHP-3 recognized Hla from *S. aureus* and CytK2 from *B. cereus* with high efficiency (Figure 5a,b and Figure 6a,b), but HlyII was recognized less efficiently. Apparently, this is due to differences in the accessibility of the epitope for this antibody in the 3D structures of the compared PFTs. Additionally, *B. cereus* HlyII contains a C-terminal domain, which can also reduce the availability of epitopes for antibodies against the HP. This assumption was tested using the HlyII∆CTD (HlyII Asp32-Leu318) protein [21], in which the HlyII C-terminal domain (Asp319-Ile412) was deleted. All HP-series mAbs recognized this protein (Figure 5a). HP-5 and the SHP-series mAbs recognized the HlyII∆CTD protein better than the full-size HlyII protein in ELISA.

Since the SHP contained the amino acid residues (Met225 and Lys226) included in HlyIILCTD, the antibodies bound to HlyIILCTD to a small extent, whereas SHP-2 interacted with HlyIILCTD to a greater extent and stained the corresponding band in the immunoblotting assay. Apparently, these amino acid residues are included in the epitopes of the SHP-series mAbs.

### 2.4. Suppression of the Cytolytic Activity of the HP-Series mAbs

The HP-5 mAb is capable of recognizing HlyII, CytK2 of *B. cereus*, and Hla of *S. aureus*. This antibody was used to investigate the possibility of inhibiting their cytolytic activity. Analysis showed that HP-5 significantly reduced the cytolytic activity of CytK2 but did not affect the activity of Hla or HlyII (Figure 7a). A three-dimensional structural model of CytK2 and HlyII was obtained based on X-ray diffraction analysis of Hla (Figure 4a). It is possible that slight variations in the three-dimensional structures are enough to alter the effectiveness of PFTs’ cytolytic activity suppression. In addition, HlyII contains a C-terminal domain, which can change the availability of individual amino acids for mAbs. Using HlyII∆CTD, which lacks the C-terminal domain [21], we confirmed the effect of HlyIICTD on the suppression of HlyII hemolytic activity by HP-5 antibodies. HP-5 suppressed the cytolytic activity of HlyII∆CTD (Figure 7b). The protective kinetics against hemolysis of rabbit erythrocytes exposed to CytK2 and HlyII∆CTD, depending on the number of PFTs and the amount of HP-5, are shown in Figure 7.

## 3. Discussion

Bioinformatic analysis of β-PFTs revealed, in their amino acid sequences, a high-homology region that corresponds to the Lys171-Gly250 region of *B. cereus* HlyII. This HP contains the linear sequence YGNQLFM, identical to certain PFTs.

Production of mAbs against the HP and SHP (213DSFNTFYGNQLFMK226) peptides confirmed the conclusion made in the bioinformatic analysis regarding a high-homology region (including an identical section) in the primary sequences of β-pore-forming toxins. All obtained antibodies that simultaneously recognized various toxins belonged to class M—early immune response antibodies—which may indicate a high level of adaptation of pathogens to overcome the immune defense of the “host” and to weaken the formation of long-term immune memory.

All antibodies that simultaneously recognized the toxins under study recognized monomeric water-soluble forms of toxins; therefore, their close epitopes are located on the surface of the proteins. Figure 8 shows the location of the binding regions of the mAbs obtained against the HP and SHP on the 3D model of the HlyII monomer, predicted by AlphaFold [18,19]. HP-4 recognizes the region including HlyIILCTD [15]. The HP-1, HP-3, and HP-7 mAbs recognize the HlyII regions located upstream from the identical section. The experimental data demonstrate that HP-5 recognizes a region containing an identical section, as confirmed by the production of mAbs against the SHP DSFNTFYGNQLFMK, since the mAbs of the SHP series obtained against the SHP containing an identical region also simultaneously recognized HlyII, CytK2, and Hla.

The amino acid sequence of the CytK2 protein is 37% identical to the sequence of *B. cereus* hemolysin II, and 30% identical to the sequence of *S. aureus* α-hemolysin [22], which suggests that, from a structural point of view, CytK2, like HlyII, belongs to the family of oligomeric β-barrel PFTs [23]. It should be borne in mind that there are no experimental data on the 3D structural models of either HlyII and CytK2, and the model of their spatial structure [24] was obtained on the basis of a previously constructed model based on X-ray diffraction analysis of Hla [17]. In this regard, it can be assumed that the accessibility of individual amino acids on the surface of the three PFTs studied may be different for mAbs. These differences likely determine the efficiency of interactions with mAbs. The suppression of the cytolytic activity of CytK2, as well as the inability of antibody HP-5 to suppress the cytolytic activity of HlyII and Hla, is possibly determined by some differences in the 3D structures of these PFTs, which are significant for the effect of HP-5 on the hemolytic activity of the toxins.

HlyII, unlike Hla and CytK2, contains a C-terminal domain. It is known that deletion of the CTD leads to an eightfold decrease in hemolytic activity towards rabbit erythrocytes when compared with intact HlyII [24]. The work [25] describes a model in which the core part and HlyIICTD are close in the full-sized HlyII. Therefore, antibody binding sites can be shielded in the full-length toxin. It is also possible to change the 3D structure of the full-length protein in the absence of HlyIICTD. The use of a deletion variant of HlyII lacking HlyIICTD revealed a marked suppression of the cytolytic activity of HlyII∆CTD by the HP-5 antibody.

A decrease in the efficiency of the recognition and suppression of cytolytic activity may indicate that the epitopes located on HlyII for all antibodies obtained in monomeric water-soluble form may be partially altered under the influence of HlyIICTD. Previously, the authors showed that HlyIICTD is capable of binding to cell membranes, oligomerizing to form ion channels. It has been shown that during the oligomerization of HlyIICTD, conformational changes in its antigenic structure occur, which manifests itself in a change in interaction with monoclonal antibodies [26]. Moreover, the ability to recognize all three of the PFTs analyzed suggests that the HP-5 and SHP-series mAbs will be able to recognize other β-barrel PFTs.

The SHP-series mAbs, similarly to the SHP, are able to recognize CytK2, Hla, and HlyII, as is the HP-5 monoclonal antibody. In addition, they show a similar effect on toxins: they suppress CytK2 activity and do not affect Hla. These data allow us to conclude that the epitope for HP-5 and the epitopes for the SHP-series mAbs are located at the same site of the HP. Thus, the localization of epitopes for the analyzed monoclonal antibodies in the sequence of the HP is as follows: HP-1, HP-3, and HP-7 bind to HlyII within Lys171-Arg212, HP-4 binds to the epitope within Met225-Gly250; HP-5, as well as the SHP-series antibodies, bind to the epitope within 213-DSFNTFYGNQLF-224, while apparently the Phe215-Phe218 site is not essential for binding these antibodies, since FNTF is located in this place in HlyII, YHSL in CytK2, and WNPV in Hla, but can affect the function of suppressing hemolytic activity by antibodies. All three PFTs examined in this work were recognized by the HP-5 mAb. However, this antibody suppressed the hemolytic activity of only CytK2 and HlyIIΔCTD, in contrast to Hla. Comparison of the amino acid sequences of the homologous peptide region of the three PFTs revealed that Hla was enriched in proline residues (five residues in the homologous peptide section). One of them is located in the HP-5 region sequence instead of Tre217 in HlyII and Ser236 in CytK2, while CytK2 and HlyIIΔCTD each have two proline residues outside this region (213-DSFNTFYGNQLF-224). Since the obtained mAbs interact with native proteins, which confirms their functional activity, their epitopes are located on the surface of the spatial structure. The presence or absence of a proline residue capable of changing the orientation of the polypeptide chain is essential for the formation of a 3D structure and, therefore, conformational epitopes. We previously demonstrated the role of proline in the accessibility of the amino acid sequences of mAbs. It was shown in [11,27] that the binding of the antibody against the C-terminal domain with full-sized toxins characterized by reciprocal mutations and L324P P324L differs, while the inhibition of hemolytic activity by this antibody differs to a greater extent. It was shown in [27] that substitutions of proline residues at positions 324 and 405 also led to a significant change in the affinity of antibodies to the C-terminal domain. The effect of P405 on the 3D structure of HlyIICTD was described in [28,29]. Thus, the presence of proline residues can significantly change the 3D structure of proteins and can modulate the suppression of the activity of PFTs.

## 4. Materials and Methods

### 4.1. Strains, Plasmid, and Enzymes

The following *E. coli* strains were used: XL1-Blue *recA1 endA1 gyrA96 thi-1 hsdR17 supE44 relA1 lac [F’proAB lacIqZΔM15 Tn10 (Tetr)]* for gene cloning and BL21(DE3) F–*ompT hsdSB* (rB–, mB–) *gal dcm* (DE3) for high-level T7 expression of recombinant proteins.

Restriction endonucleases KpnI and NdeI (Thermo Scientific, Waltham, MA, USA), T4-DNA ligase (NEB, Ipswich, MA, USA), protein markers and DNA electrophoresis markers (Thermo Scientific, Vilnius, Lithuania), Q5 High-Fidelity DNA polymerase (NEB, Ipswich, MA, USA), TaqSE-DNA polymerase (SibEnzyme, Moscow, Russia), and dNTP mix (Thermo Scientific, Waltham, MA, USA) were used. The PCR product was cloned using the pET29b (+) vector plasmid.

### 4.2. Molecular Cloning

Plasmids pET29-hlyIIΔSP14579, pET29-hlyIIΔSPΔCTD14579, and pET28-hlaΔSP encoding intracellular HlyII, HlyIIΔCTD, and Hla, respectively, were obtained earlier [11,21].

To create the pET29-HP plasmid, the gene encoding the homologous peptide (Lys171-Gly250 region of *B. cereus* HlyII) was amplified from genomic DNA using the following primers:

HlyIIfrgomF: 5’-CCTCTAGACATATGAAAGAAAGTGTATCTTATGATC

HlyIIfrgomR: 5’-CTCGAGGGTACCACCATATCCTGTTAAAGC

Similarly, to create the pET29-cytK2ΔSP plasmid, the gene encoding intracellular CytK2 was amplified from genomic DNA using the following primers:

F-CytK2ΔSP: 5’-TTATAGGATCCCATATGCAAACGACGTCACAAG

R-CytK2ΔSP: 5’-TTACTCGAGGGTACCTTTTTTCTCTACCAATTTCTTATTC

The PCR products were cloned into the pET29b vector using NdeI and KpnI restriction enzymes. This ensured that the final product included six histidine residues and a thrombin recognition site. This allows for the removal of the six histidines if necessary using the thrombin enzyme.

To produce proteins, the *E. coli* BL21(DE3) strain was transformed with recombinant plasmid.

### 4.3. Protein Expression and Purification

The expression of all protein products followed the same procedure. The culture was grown at 37 °C in LB medium in two flasks (200 mL each), containing kanamycin at a concentration of 20 μg/mL, until the optical density at 600 nm (A_600_) reached 0.7–0.9. Protein expression was then induced by adding isopropyl β-D-thiogalactoside (IPTG) to a final concentration of 0.1 mM, and the cultivation was continued at 20 °C for 12 h. The biomass was harvested by centrifugation at 6000× *g* for 15 min. The pellet was resuspended in 20 mL of buffer A (50 mM sodium hydrogen phosphate, 500 mM sodium chloride, 5% glycerol, pH 8.0), and treated with 1 mM phenylmethylsulfonyl fluoride (PMSF) and 0.1% lysozyme. The resulting cell suspension was sonicated using a QSonica Q700 (QSonica, Newtown, CT, USA) ultrasonic homogenizer for 7 cycles of 20 s each, with 2 min breaks between cycles at an amplitude of 40%. After sonication, the disrupted cells were centrifuged using a Beckman Coulter Avanti JXN-26 (Beckman Coulter, Brea, CA, USA) centrifuge at 16,000× *g* for 60 min at 6 °C. The clarified lysate and remaining cell debris were then analyzed by SDS-PAGE. For further purification, the cell debris was washed with 10 mL of buffer B (100 mM NaH_2_PO_4_, 10 mM Tris-HCl, pH 8) containing 2 M urea. The debris was then dissolved in another solution of buffer B with 8 M urea and purified using Ni-NTA metal chelate affinity chromatography (Qiagen, Germantown, TN, USA) according to the manufacturer’s protocol. The purified denatured homologous peptide was used to immunize mice. The other proteins were refolded on Ni-NTA in the presence of 0.125 M arginine and used for ELISA and suppression of hemolytic activity by the mAbs.

### 4.4. Preparation and Purification of mAbs

For immunization in the production of mAb-secreting hybridomas, we used animals kept under standard conditions in accordance with the Decree of the Chief State Sanitary Doctor of the Russian Federation dated 29 August 2014, No. 51 “On approval of SR 2.2.1 “Maintenance of experimental biological clinics (vivariums)”. For immunization, a homogeneous preparation of the homologous peptide in 8 M urea was used.

Two groups of BALB/c mice (5 females, 2–3 months of age) were immunized. One batch of mice was immunized with a dose of 15 μg of homologous peptide/mouse; the other was immunized with a dose of 30 μg/mouse in Freund’s complete adjuvant. Animals of the first group were further immunized with a dose of 10 μg/mL in Freund’s incomplete adjuvant. The second group was further immunized with a dose of 20 μg/mL, also in Freund’s incomplete adjuvant. Five immunizations were carried out at two-week intervals.

SHP conjugation to KLH (Merck, Darmstadt, Germany) was carried out using 0.1% glutaraldehyde as a cross-linking agent. Before the conjugation procedure, 50 mg of KLH was suspended in 1 mL of water, sonicated and stirred for 4 hours at 40 °C, then dialyzed against 0.1 M na-phosphate buffer, pH 7.8, overnight at 4 °C and centrifuged at 10,000× *g*. Then, 5 μg each of the peptide and glutaraldehyde were added to the KLH solution. Conjugation was carried out for 12 h with stirring, with pH control, and, if necessary, neutralization with NaOH.

Immunization of the experimental animals was performed with a dose of 10 μg of peptide per 100 μg of KLH in Freund’s adjuvant according to the scheme used for immunization with the homologous peptide.

The mAb-secreting hybridomas were obtained using hybridoma technology [20]. Selection of hybridomas secreting specific antibodies was carried out via indirect ELISA using the interaction of supracellular supernatants with HlyII and Hla immobilized on immunoplates.

The mAbs obtained were preparatively produced in the culture liquid during the cultivation of mAb-secreting hybridomas. Class G mAbs were purified by protein A-sepharose affinity chromatography [30]. IgM was purified by 3-fold precipitation with 50% ammonium sulfate saturation.

The types of heavy and light chains in immunoglobulin were determined using ELISA, specifically a Rapid ELISA Mouse mAb Isotyping Kit (Thermo Fisher Scientific, Waltham, MA, USA) according to the manufacturer’s instructions.

### 4.5. Determination of the Protein and Peptide Concentrations

To determine the antibody concentration, an extinction weight coefficient of 1.185 was used [31]. Protein concentrations were determined using spectrophotometry, taking into account their extinction coefficients calculated from the amino acid sequence using the ExPASy ProtParam tool (accessed on 15 Jan 2024) [32]. 

### 4.6. The Enzyme-Linked Immunosorbent Assay

The antigens (*B. cereus* CytK2, HlyIILCTD, HlyII, HlyII∆CTD, and *S. aureus* Hla) were added to the wells of the ELISA plates at a concentration of 1 μg/mL in buffer (0.05 M carbonate buffer, pH 9.6) overnight at +4 °C. To block possible nonspecific binding, the free plastic binding centers were blocked for 30 min with PBST (phosphate-buffered saline containing 0.1% Tween-20). The mAbs were added to the experimental wells at a concentration of 5 μg/mL and incubated for 1 h at 37 °C. The resulting antigen–antibody complexes were detected with goat anti-mouse immunoglobulin conjugate with horseradish peroxidase (ThermoScientific, Waltham, MA, USA) in PBST diluted according to the manufacturer’s instructions. After incubation with each reagent, the experimental wells were washed at least six times with excess PBST. In the last stage, a 4 mM solution of *ortho*-phenylenediamine (Sigma, St. Louis, MO, USA) in citrate–phosphate buffer (26 mM citric acid, 50 mM Na_2_HPO_4_, pH 5.0) containing 0.003% H_2_O_2_ (*v*/*v*) was added to the wells of the immunoplate. The reaction was stopped by adding an equal volume of 10% (*v*/*v*) sulfuric acid. Optical absorption at 490 nm was determined using an iMark photometer (Bio-Rad, Berkeley, CA, USA). The A_490_ value measured in wells that did not have antibodies added, corresponding to the nonspecific binding of the conjugate to the adsorbed antigen (negative control), was subtracted from the A_490_ value in wells to which antibodies were added.

### 4.7. Immunoblotting

Electrophoretic separation of the studied proteins was carried out in a 14% polyacrylamide gel according to [33]. Proteins were added to the pockets of the concentrating gel at 0.4 μg. Transfer to a nitrocellulose membrane and staining of protein bands were carried out as described in [21], with the mAbs added at a concentration of 10 μg/mL.

### 4.8. PFT Hemolytic Activity Suppression with the mAbs

To evaluate the potential for suppression of hemolytic activity, different concentrations of the investigated PFTs were prepared by two-fold serial dilutions and incubated with the mAbs at a concentration of 0.4 μM for 15 min at 37 °C. To determine the effect of the concentration of the mAbs on the suppression of hemolytic activity, concentrations of mAbs ranging from 0.4 to 0.003 μM, obtained by two-fold serial dilutions, were incubated with a specific amount of toxin (see Figure 6). The incubation was carried out at 37 °C for 15 min. An equal volume of a 1% rabbit erythrocyte suspension was then added, and the mixture was further incubated for 30 min at the same temperature. Following the incubation, the erythrocytes were pelleted by centrifugation, and the supernatant was used for spectrophotometric analysis of the hemolysis levels via optical density measurement at a wavelength of 541 nm.

To construct normalized hemolytic activity curves, all PFT concentrations were converted to hemolytic units (HUs). A HU is defined as the minimum quantity of toxin that causes hemolysis in 50% of a 1% suspension of rabbit erythrocytes after a 30 min incubation at 37 °C. The hemolysis level is expressed as a percentage of hemolysis, where the maximum value of the optical density of the supernatant after the hemolysis reaction, measured at a wavelength of 541 nm, was taken as 100%.

### 4.9. Statistical Analysis

The analysis was performed using Microsoft Excel and STATISTICA v. 7.1. The Mann–Whitney test was used to analyze the statistical significance between the experimental groups. Statistical significance was achieved at *p* < 0.05. Data are presented as means ± standard deviations (SDs).

## 5. Conclusions

In this work, bioinformatic analysis revealed a high-homology region with an identical section YGNQLFM in the sequences of β-pore-forming toxins. These data were confirmed by obtaining monoclonal antibodies that simultaneously recognized HlyII, CytK2 of *B. cereus*, and Hla of *S. aureus*. It has been shown that antibodies that recognize an identical site are able to suppress the hemolytic activity of HlyII∆CTD and CytK2.

## Figures and Tables

**Figure 1 ijms-25-05327-f001:**
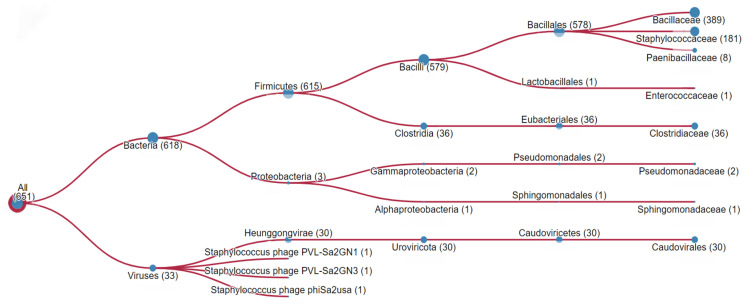
The taxonomic distribution of significant hits with an E-value less than 0.003 from the search for the HlyII Lys171-Gly250 sequence (HP) using the HMMER [12] tool in the UniProtKB [12,13] database.

**Figure 2 ijms-25-05327-f002:**
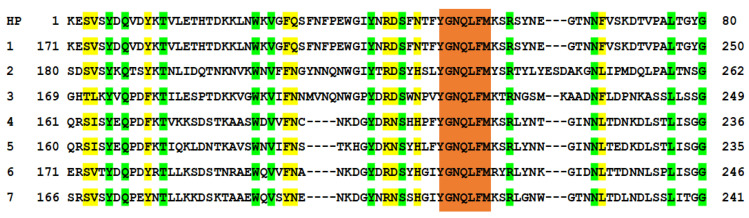
Comparison of the HlyII Lys171-Gly250 sequence with the β-barrel PFT sequences. 1—Hemolysin II of *B. cereus*; 2—cytotoxin K2 of *B. cereus*; 3—α-hemolysin of *S. aureus*; 4—necrotizing enteritis toxin NetG of *C. perfringens;* 5—leukotoxin domain protein A of *C. perfringens*; 6—β-channel forming cytolysin of *C. septicum*; 7—leukocidin family pore-forming toxin of *C. botulinum*. Positions with identical amino acid residues are highlighted in green. Positions where similar properties among groups of amino acids are conserved are marked in yellow. Orange refers to an identical section.

**Figure 3 ijms-25-05327-f003:**
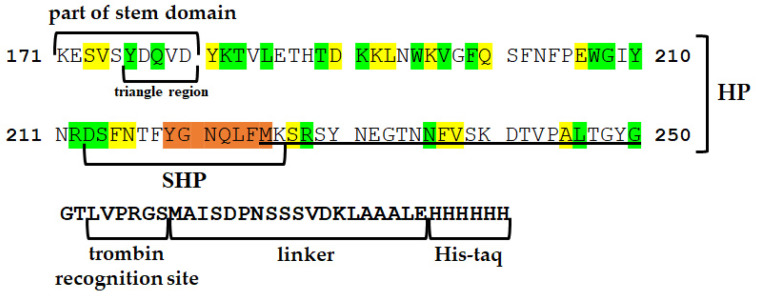
Amino acid sequence of the cloned HP against which HP-series mAbs were obtained. The amino acids are numbered in relation to the sequence of the full-length hemolysin II. Positions with identical amino acid residues for Hla, CytK2, and HlyII are shown in green. Yellow indicates positions where similar properties between groups of amino acids are preserved. Orange refers to an identical section. The sequence including the HlyIILCTD [15] region is underlined. The thrombin recognition site, linker, and six histidine residues are highlighted in bold. For the synthetic peptide against which the SHP-series mAbs were obtained, part of the stem domain and triangle region [14] are noted with signatures.

**Figure 4 ijms-25-05327-f004:**
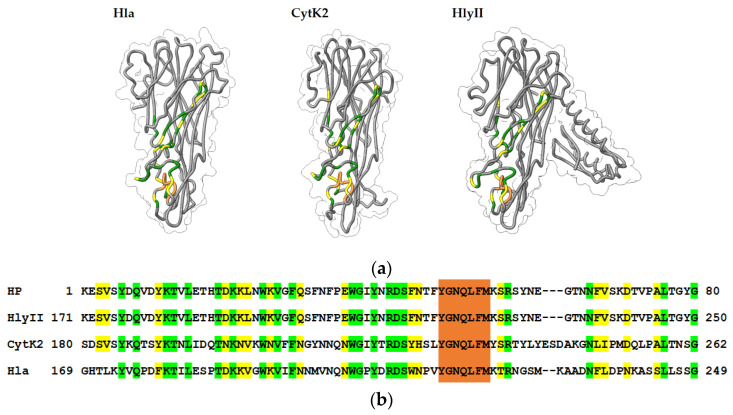
(**a**) Location of the homologous regions on models of the water-soluble monomer proteins Hla (Uniprot: Q2G1X0), CytK2 (Uniprot: Q81GS6), and HlyII (Uniprot: Q81AN8), with structures predicted by AlphaFold [18,19]. Positions with identical amino acid residues for Hla, CytK2, and HlyII are shown in green. Yellow indicates the homologous amino acids of the three selected PFTs. The identical section is highlighted in orange. (**b**) Homologous peptide alignment in Hla, CytK2, and HlyII. Orange refers to an identical section.

**Figure 5 ijms-25-05327-f005:**
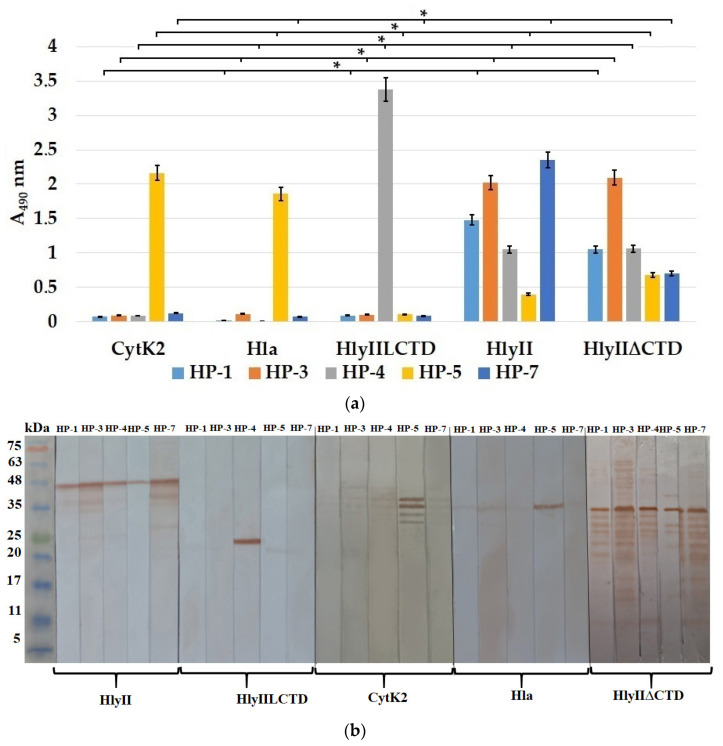
(**a**) Comparison of the interaction of the HP-series mAbs (concentration, 5 μg/mL) with *B. cereus* CytK2, *S. aureus* Hla, and HlyIILCTD, HlyII, and HlyII∆CTD of *B. cereus* during the sorption from a concentration of 1 μg/mL of toxins in ELISA. Data are represented as the means ± SDs of 5 independent repeats (*n* = 5). * A statistically significant difference (*p* < 0.05, Mann–Whitney). (**b**) Immunoblotting of HlyII, HlyIILCTD, CytK2 of *B. cereus*, Hla of *S. aureus*, and HlyII∆CTD of *B. cereus* with the mAbs against the HP. Before electrophoretic separation, the samples were boiled in 1% SDS for 10 min. The names of the mAbs are given above the lanes. The image was obtained by combining four immunoblots.

**Figure 6 ijms-25-05327-f006:**
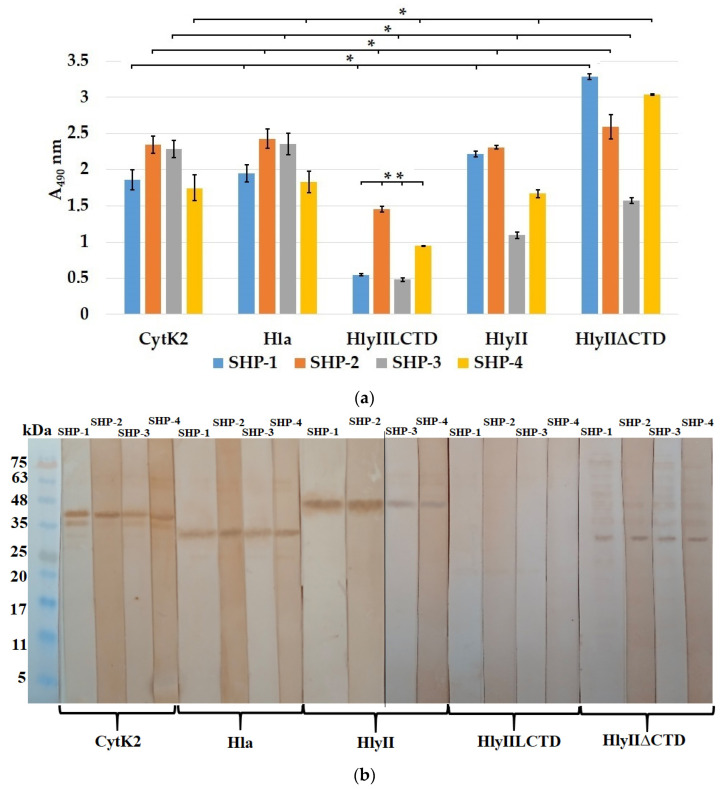
(**a**) Comparison of the interaction of the SHP-series mAbs (concentration, 5 μg/mL) with *B. cereus* CytK2, *S. aureus* Hla, and HlyIILCTD, HlyII, and HlyII∆CTD of *B. cereus* during the sorption from a concentration of 1 μg/mL of toxins in ELISA. Data are represented as the means ± SDs of 5 independent repeats (*n* = 5). * A statistically significant difference when comparing the interaction of antibodies with different antigens. (*p* < 0.05, Mann–Whitney); ** a statistically significant difference when comparing the antibody binding to HlyIILCTD (*p* < 0.05, Mann–Whitney). (**b**) Immunoblotting of CytK2 of *B. cereus,* Hla of *S. aureus*, and HlyII, HlyIILCTD, and HlyII∆CTD of *B. cereus* with the mAbs against the SHP. Before electrophoretic separation, the samples were boiled in 1% SDS for 10 min. The names of the mAbs are given above the lanes. The image was obtained by combining two immunoblots.

**Figure 7 ijms-25-05327-f007:**
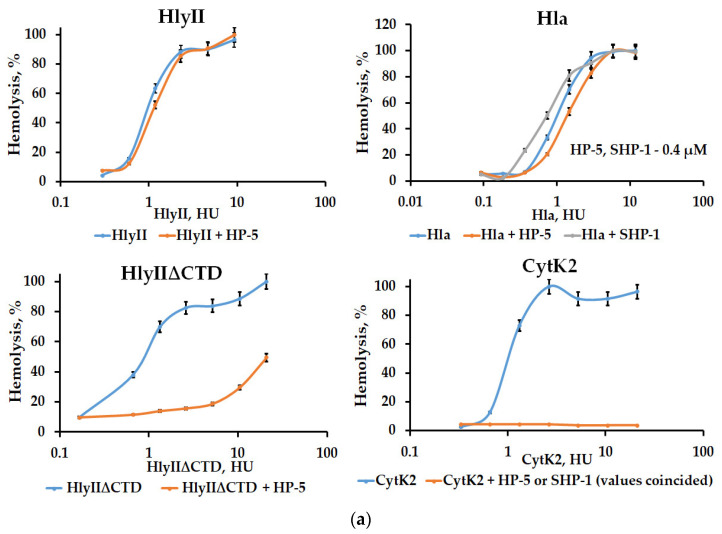
Suppression of the hemolysis of the HP-5 mAb during an attack of rabbit erythrocytes by various PFTs. (**a**) Hemolytic activity of PFTs without the addition of the mAb (blue line) and after the addition of the mAb to a final concentration of 0.4 μM (orange line). (**b**) Hemolytic activity of the PFTs after the addition of the mAb at different concentrations. The hemolytic activity curves without the addition of the mAb are shown on the left. Colored arrows indicate points on the curves corresponding to the concentrations of the pore-forming toxin incubated with the mAb. On the right, suppression of hemolysis depending on the concentration of the mAb. The color of the arrows corresponds to the color of the lines. Data are represented as the means ± SDs of 5 independent repeats (*n* = 5).

**Figure 8 ijms-25-05327-f008:**
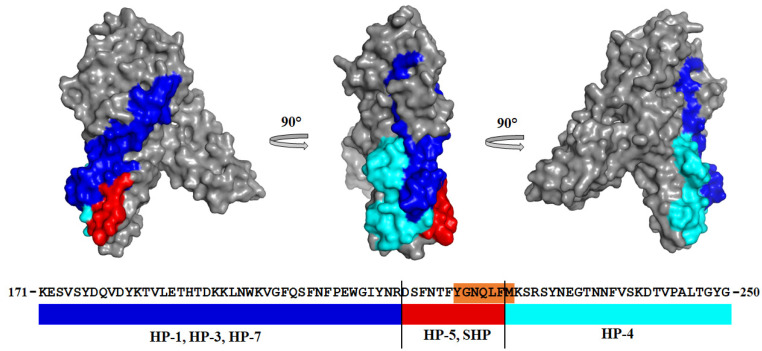
Location of putative regions containing epitopes to the HP-1, HP-3, HP-4, HP-5, and HP-7 mAbs on the HlyII (UniProt: Q81AN8) water-soluble monomer model. Blue: HP-1, HP-3, and HP-7; red: HP-5 and SHP-series mAbs (including an identical (Tyr219-Met225) section (orange)); cyan: HP-4 (includes a section of HlyIILCTD).

## Data Availability

Data is contained within the article.

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
