# Peer review of "A High-Homology Region Provides the Possibility of Detecting β-Barrel Pore-Forming Toxins from Various Bacterial Species"

_ijms, 2024, doi:10.3390/ijms25105327_

Round 1

Reviewer 1 Report

Comments and Suggestions for Authors

The manuscript submitted by Nagel and colleagues titled “A High-Homology Region in Bacterial β-Barrel Pore-Forming Toxins” uncovers an “homologous peptide” shared among several PFTs. The objective of this work is to generate antibodies against this peptide as a tool for the identification of β-Barrel PFTs across different species and for the potential inhibition of the cytolytic activity of selected β-Barrel PFTs. Through a combination of bioinformatics analysis, molecular biology and biochemistry, Nagel et al., demonstrate that a monoclonal antibody (GP-5) targeting the epitope within the "homologous peptide" successfully detects the three PFTs used in this work. Furthermore, this GP-5 antibody effectively inhibits the hemolytic activity of CytK2 and HlyIIΔCTD. These findings carry substantial implications and may prove valuable for studying other PFTs. However, the manuscript needs to be considerably improved and several major points require clarification in order to be considered for publication (see below).

 Major Points:

·   In the Abstract/Introduction, the authors designate as “homologous peptide” the region homologous to the Lys171-Gly250 sequence in HlyII from B. cereus, however, in the results section, it is not clear what is the “homologous peptide”. For example, in the Figure 2 the authors show an alignment of the “homologous peptide” (Lys171-Gly250) but the title of this figure (line 96), is “Conserved regions of part of a β-barrel PFT including the homologous peptide”. This title suggests that the “homologous peptide” is part of the conserved regions Lys171-Gly250, what is not in agreement with the definition of the “homologous peptide” gave by the authors in the abstract/introduction. The authors should clarify this in all the text (for example, indicating the specific residues that compose the referring peptide). Moreover, it is not clear what “GP” designation means, if is the “homologous peptide” or the antibodies (Line 116: “which mAbs designated as GP were obtained.”). Also, to be clearer, I suggest the authors to define the homologous peptide cloned as “HP” and the synthetic homologous peptide as “SHP” for example, in all the text.

·   Line 184-187: The authors suggest that the C-terminal domain (CTD) of HlyII can reduce the accessibility of the epitope for the GP-5 antibody, but interestingly, the CTD is on the opposite side to the site to which the mAb binds. It is not very well understood how the presence of the domain prevents the binding of the Ab to its epitope. The results in figure 5 showing a “high efficiency” of GP-5 for CytK2 and Hla are not replicated in HlyIIΔCTD, which presents a slight increase in the interaction with GP-5 when compared with HlyII. Together, this observations do not fully justify the argument presented by the authors. Do you know what is the function of this domain? Promoting the oligomerization of the toxin and, in the oligomeric state, preventing the binding of the mAb to the peptide/epitope?

·   Line 232: The authors state “Production of mAbs against homologous and synthetic (DSFNTFYGNQLFMK) peptides confirmed…”, but the synthetic peptide is also homologous. The authors need to find another nomenclature to make the text more understandable and correct.

·   Figure 3a: The authors should indicate the stem and triangle region and the residues number on the amino acid sequence to facilitate the readers. Moreover, in the legend of the Figure 3, the authors state “The recombinant part of the protein sequence is highlighted in bold.”, but the rest of the sequence is also a recombinant protein. Please clarify the legend.

·   Structure models: The authors should explain how the models were done (AlphaFold, Swiss-Model,…) and indicate the PDB code for Hla.

·  It will be easier for the readers if the authors indicated the region corresponding to the proteins HlyIILCTD (M225-I412?) and HlyIIΔCTD (M1-L318?) instead of having to search through previous publications.

·   The authors conclude that “the primary YGNQLFM sequence identical in these PFTs, provides a tool (mAbs) for the simultaneous identification of at least three β-pore-forming toxins.” but this conclusion is based on results from HP-series mAbs that the authors don’t show. Its essential to show this results in the manuscript in order to be able to draw this conclusion.

·   Figure 5a: Deviation to the mean values should be presented as standard deviation (SD) and not SEM.

·  Figure 5b: The authors show the interactions of GP-series mAbs with HlyIIΔCTD in ELISA but not in the immunoblot. The authors should add this results in the Figure 5b.

·   Line 279-281: The authors state “while apparently the Phe215-Phe218 site is not essential for binding these antibodies – since FNTF is located in this place in HlyII; YHSL, in CytK2; and WNPV, in Hla – but can affect the function of suppressing hemolytic activity by antibodies.” To validate this hypothesis, I suggest the authors to mutate this residues YHSL in CytK2 and see if the mAbs stop inhibiting CytK2 cytolytic activity.

·   Line 284-286: What means “homologous peptide section”, Lys171-Gly250? If yes, Hla was enriched in 5 proline residues (P177, P186, P207, P215 and P239) and not 4 as the authors indicated. Moreover, CytK2 and HlyIIΔCTD each have two proline residues outside this region and not one as mentioned by the authors.

·   Line 289-290: The authors suggest that the proline residues can modulate the suppression of PFTs activity but then, shouldn't they also inhibit binding in ELISA? This affirmation should be clarified.

·   In the Materials and Methods, I recommend adding a section explaining how the authors quantified protein/antibody/peptide. The authors should describe briefly the method mentioned in the section 4.5 and 4.6 and also how the ELISA was performed.

·   The quality of the graphs should be improved, and I suggest the use of specific software for graphical design and statistical analysis. A section referring to the graphical presentation and analysis should be added in the Materials & Methods section.

 Minor Points:

 ·        Line 46: Delete “of” in the sentence “that directly damage of structures or…”

·        Line 70: Substitute “We believe” by “We hypothesize”

·     Line 81 and Line 123: “varying degrees of homology.” There are distinct degrees of similarity and identity but not homology, either they are or they are not homologous. Please change the sentences.

·        Line 84: Replace “homologuos” by homologous

·     Figure 4: In the legend, add “Uniprot” before number code (Ex: Uniprot Q2G1X0)

·        Figure 6a:  Hla in the title of the corresponding graph is missing.

·  Figure 6: In the legend, the indication of number of independent experiments (n) performed, mean values and standard deviation, are missing.

·        Line 336: Replace “Fphenylmethylsulfonyl” by Phenylmethylsulfonyl

·        Line 340: Replace “12,000 rpm” with the equivalent in g

Comments on the Quality of English Language

The manuscript is clearly written, particularly the abstract and the introduction sections, making it accessible to readers who are not familiar with this specific topic and is aligned with the scope of IJMS. However, the results and discussion sections present several statements that cause confusion and require a considerable reformulation (see major points).

Author Response

The authors are grateful to the reviewer for high opinion, carefully reading the manuscript and for comments.

 Major Points:

  •  In the Abstract/Introduction, the authors designate as “homologous peptide” the region homologous to the Lys171-Gly250 sequence in HlyII from B. cereus, however, in the results section, it is not clear what is the “homologous peptide”. For example, in the Figure 2 the authors show an alignment of the “homologous peptide” (Lys171-Gly250) but the title of this figure (line 96), is “Conserved regions of part of a β-barrel PFT including the homologous peptide”. This title suggests that the “homologous peptide” is part of the conserved regions Lys171-Gly250, what is not in agreement with the definition of the “homologous peptide” gave by the authors in the abstract/introduction. The authors should clarify this in all the text (for example, indicating the specific residues that compose the referring peptide). Moreover, it is not clear what “GP” designation means, if is the “homologous peptide” or the antibodies (Line 116: “which mAbs designated as GP were obtained.”). Also, to be clearer, I suggest the authors to define the homologous peptide cloned as “HP” and the synthetic homologous peptide as “SHP” for example, in all the text.

The authors took into account your comments and made changes to the text. The manuscript clarified the numbering of amino acid residues for each of the proteins and peptides used, as well as changed the nomenclature of peptides and antibodies.

  •  Line 184-187: The authors suggest that the C-terminal domain (CTD) of HlyII can reduce the accessibility of the epitope for the GP-5 antibody, but interestingly, the CTD is on the opposite side to the site to which the mAb binds. It is not very well understood how the presence of the domain prevents the binding of the Ab to its epitope. The results in figure 5 showing a “high efficiency” of GP-5 for CytK2 and Hla are not replicated in HlyIIΔCTD, which presents a slight increase in the interaction with GP-5 when compared with HlyII. Together, this observations do not fully justify the argument presented by the authors. Do you know what is the function of this domain? Promoting the oligomerization of the toxin and, in the oligomeric state, preventing the binding of the mAb to the peptide/epitope?

In the text of the manuscript (discussion section), the authors tried their best to explain their concept of possible reasons for changes in the interaction of antibodies with full-sized HlyII and HlyIIΔCTD.

  •  Line 232: The authors state “Production of mAbs against homologous and synthetic (DSFNTFYGNQLFMK) peptides confirmed…”, but the synthetic peptide is also homologous. The authors need to find another nomenclature to make the text more understandable and correct.

Nomenclature changed

  •  Figure 3a: The authors should indicate the stem and triangle region and the residues number on the amino acid sequence to facilitate the readers. Moreover, in the legend of the Figure 3, the authors state “The recombinant part of the protein sequence is highlighted in bold.”, but the rest of the sequence is also a recombinant protein. Please clarify the legend.

Clarifying changes have been made to the figure and legend.

  •  Structure models: The authors should explain how the models were done (AlphaFold, Swiss-Model,…) and indicate the PDB code for Hla.

Appropriate explanations have been made

  • It will be easier for the readers if the authors indicated the region corresponding to the proteins HlyIILCTD (M225-I412?) and HlyIIΔCTD (M1-L318?) instead of having to search through previous publications.

The regions of the amino acid sequence HlyII corresponding to the proteins HlyIILCTD and HlyIIΔCTD are included in the text of the manuscript

  •  The authors conclude that “the primary YGNQLFM sequence identical in these PFTs, provides a tool (mAbs) for the simultaneous identification of at least three β-pore-forming toxins.” but this conclusion is based on results from HP-series mAbs that the authors don’t show. Its essential to show this results in the manuscript in order to be able to draw this conclusion.

The results of the binding of antibodies to the studied antigens of the SHP-series, obtained against a synthetic peptide, are given in the manuscript

  •  Figure 5a: Deviation to the mean values should be presented as standard deviation (SD) and not SEM.

Corrected

  • Figure 5b: The authors show the interactions of GP-series mAbs with HlyIIΔCTD in ELISA but not in the immunoblot. The authors should add this results in the Figure 5b.

Results added

  •  Line 279-281: The authors state “while apparently the Phe215-Phe218 site is not essential for binding these antibodies – since FNTF is located in this place in HlyII; YHSL, in CytK2; and WNPV, in Hla – but can affect the function of suppressing hemolytic activity by antibodies.” To validate this hypothesis, I suggest the authors to mutate this residues YHSL in CytK2 and see if the mAbs stop inhibiting CytK2 cytolytic activity.

The goal of this work was to find the possibility of simultaneous detection of β-barrel pore-forming toxins. The authors express their gratitude to the reviewer and plan to carry out the experiments suggested by the reviewer. Moreover, the authors are already working in this direction. However, the authors are unable to complete these studies within the established time frame. We plan that these results will be included in our next publication.

  •  Line 284-286: What means “homologous peptide section”, Lys171-Gly250? If yes, Hla was enriched in 5 proline residues (P177, P186, P207, P215 and P239) and not 4 as the authors indicated. Moreover, CytK2 and HlyIIΔCTD each have two proline residues outside this region and not one as mentioned by the authors.

Text has been corrected

  •  Line 289-290: The authors suggest that the proline residues can modulate the suppression of PFTs activity but then, shouldn't they also inhibit binding in ELISA? This affirmation should be clarified.

The cited articles contain the necessary data, the text (discussion section) is expanded

  •  In the Materials and Methods, I recommend adding a section explaining how the authors quantified protein/antibody/peptide. The authors should describe briefly the method mentioned in the section 4.5 and 4.6 and also how the ELISA was performed.

The experimental part of the manuscript has been expanded

  •  The quality of the graphs should be improved, and I suggest the use of specific software for graphical design and statistical analysis. A section referring to the graphical presentation and analysis should be added in the Materials & Methods section.

Graphics have been improved, the corresponding section has been added

 Minor Points:

  • Line 46: Delete “of” in the sentence “that directly damage of structures or…”
  • Line 70: Substitute “We believe” by “We hypothesize”
  •    Line 81 and Line 123: “varying degrees of homology.” There are distinct degrees of similarity and identity but not homology, either they are or they are not homologous. Please change the sentences.
  • Line 84: Replace “homologuos” by homologous
  • Figure 4: In the legend, add “Uniprot” before number code (Ex: Uniprot Q2G1X0)
  • Figure 6a:  Hla in the title of the corresponding graph is missing.
  • Figure 6: In the legend, the indication of number of independent experiments (n) performed, mean values and standard deviation, are missing.
  • Line 336: Replace “Fphenylmethylsulfonyl” by Phenylmethylsulfonyl
  • Line 340: Replace “12,000 rpm” with the equivalent in g

All comments have been corrected

Reviewer 2 Report

Comments and Suggestions for Authors

The manuscript by Nagel et al. discusses the generation and characterization of monoclonal antibodies against a conserved region in bacterial pore-forming toxins. Two sets of antigens were used to generate GP-series and HP-series antibodies. GP-series antibodies were then subjected to ELISA and immunoblot analysis to elucidate their ability to bind the various PFTs.  Finally, the ability of the mAbs to suppress the PFTs was tested using a rabbit erythrocyte suspension. Overall the manuscript is very interesting and has done a wonderful job. However, the following comments need to be addressed to improve the clarity of the manuscript.

1) GP series antibodies are extensively described but the HP series antibodies are not discussed well. Either include the ELISA and immunoblot results for the HP series antibody or just drop them from the manuscript altogether.

2) Just referring to a paper for a whole methods section is not acceptable. Provide details on how it was done for your particular set of experiments

3) In Figure 4 and Figure 7, please list the PDBs used for generating the figure. Furthermore, change the ball and stick representation in Figure 4 to a ribbon model.

4) How were the potential epitopes for the various antibodies determined?

5) The title of the manuscript does not accurately reflect the work done. So please change it.

Comments on the Quality of English Language

The sentence structure and tenses are wrong in several places in the manuscript. Please take a closer look and fix them. This is not a major issue and does not diminish the impact of the paper but will help to improve the clarity of the manuscript.

Author Response

The authors are grateful to the reviewer for high opinion, carefully reading the manuscript and for comments.

1) GP series antibodies are extensively described but the HP series antibodies are not discussed well. Either include the ELISA and immunoblot results for the HP series antibody or just drop them from the manuscript altogether.

The results of the binding of antibodies to the studied antigens of the SHP-series, obtained against a synthetic peptide, are given in the manuscript. Nomenclature changed.

2) Just referring to a paper for a whole methods section is not acceptable. Provide details on how it was done for your particular set of experiments

The experimental part of the manuscript has been expanded.

3) In Figure 4 and Figure 7, please list the PDBs used for generating the figure. Furthermore, change the ball and stick representation in Figure 4 to a ribbon model.

Figures and legends corrected.

4) How were the potential epitopes for the various antibodies determined?

The authors did not define epitopes in the strict sense of the word; regions were identified in which the epitopes recognized by the resulting antibodies were included. Clarifying details have been added to the text.

5) The title of the manuscript does not accurately reflect the work done. So please change it.

The title of the manuscript has been changed.

Round 2

Reviewer 1 Report

Comments and Suggestions for Authors

The authors have adequately addressed all of my comments and incorporated corresponding revisions into the manuscript.

I endorse the publication of this article.

Reviewer 2 Report

Comments and Suggestions for Authors

There are no issues with the revised manuscript.